# A Routing Optimization Method for Software-Defined Optical Transport Networks Based on Ensembles and Reinforcement Learning

**DOI:** 10.3390/s22218139

**Published:** 2022-10-24

**Authors:** Junyan Chen, Wei Xiao, Xinmei Li, Yang Zheng, Xuefeng Huang, Danli Huang, Min Wang

**Affiliations:** 1School of Computer Science and Information Security, Guilin University of Electronic Technology, Guilin 541004, China; 2School of Computer Science and Engineering, Northeastern University, Shenyang 110819, China; 3Institute of Automation, Chinese Academy of Sciences, Beijing 100190, China

**Keywords:** optical transport network, software-defined networking, deep Q-network, message-passing neural network, ensemble learning

## Abstract

Optical transport networks (OTNs) are widely used in backbone- and metro-area transmission networks to increase network transmission capacity. In the OTN, it is particularly crucial to rationally allocate routes and maximize network capacities. By employing deep reinforcement learning (DRL)- and software-defined networking (SDN)-based solutions, the capacity of optical networks can be effectively increased. However, because most DRL-based routing optimization methods have low sample usage and difficulty in coping with sudden network connectivity changes, converging in software-defined OTN scenarios is challenging. Additionally, the generalization ability of these methods is weak. This paper proposes an ensembles- and message-passing neural-network-based Deep Q-Network (EMDQN) method for optical network routing optimization to address this problem. To effectively explore the environment and improve agent performance, the multiple EMDQN agents select actions based on the highest upper-confidence bounds. Furthermore, the EMDQN agent captures the network’s spatial feature information using a message passing neural network (MPNN)-based DRL policy network, which enables the DRL agent to have generalization capability. The experimental results show that the EMDQN algorithm proposed in this paper performs better in terms of convergence. EMDQN effectively improves the throughput rate and link utilization of optical networks and has better generalization capabilities.

## 1. Introduction

The optical transport network (OTN) is a transport network that enables the transmission, multiplexing, route selection, and monitoring of service signals in an optical domain, ensuring its performance index and survivability. The OTN can support the transparent transmission of customer signals, high-bandwidth multiplexing, and configuration. It also provides end-to-end connectivity and networking capabilities. With the rapid development of network communication technology, the demand for OTN networks has increased significantly in terms of the scale of information volume, demand complexity, and dynamic spatio-temporal distribution. Unlike traditional networks, the OTN can meet more network requirements due to its suitable transmission medium, which has a high transmission speed, more data transmission, and a long transmission distance.

Traditional routing design schemes manually model network demand characteristics and design routing policies in a focused way. The traditional routing protocol is designed for wired networks, with a fixed bandwidth allocation pattern and low bandwidth utilization. It cannot provide differentiated services based on the level of assistance, nor can it cope with the rapid changes in topology and link quality standards in optical network environments. Additionally, because OTN demand has complex spatio-temporal distribution fluctuations, the optimization problem of its routing is an NP-hard problem [1]. In this case, traditional network routing design schemes do not apply to the OTN.

With the development of new network architectures, such as the software-defined networking (SDN) and the maturation of deep reinforcement learning (DRL) techniques in recent years, software-defined optical transport networks (SD-OTNs) based on the SDN are gaining popularity in the industry. Recent studies have used the DRL to address SDN-related problems, such as QoS-aware secure routing for the SDN-IoT [2], SDN routing optimization problems [3], and the SDN demand control [4]. However, due to DRL agents’ lack of generalization capabilities, they do not achieve good results in new network topologies. Thus, DRL agents cannot make correct routing decisions when presented with unexplored network scenarios during the training phase. The main reason behind this phenomenon is that graphs essentially represent computer networks. In addition, traditional DRL algorithms use typical neural network (NN) architectures (e.g., fully connected convolutional neural networks), which are unsuitable for modeling information about graph structures. Due to the computational effort and high time complexity of the routing optimization problem, traditional DRL algorithms are challenging for the DRL agent to converge quickly when addressing the network routing optimization problem. Additionally, OTN network problems are incredibly complex and have high trial-and-error costs, making it difficult to implement DRL algorithms in real optical networks.

This paper proposes an ensembles- and message-passing neural-network-based deep Q-network (EMDQN) method to solve the SD-OTN routing decision problem. The message-passing neural network (MPNN) is a deep learning (DL) method based on a graph structure [5]. The MPNN contributes to learning the relationship between graph elements and their rules. In this paper, the MPNN is used to capture information about the relationship between the demand on links and network topology, which can improve the model’s generalization ability. Despite computationally complex network problems, ensemble learning has a unique advantage that can increase sample utilization. We reweigh the sample transitions based on the uncertainty estimates of ensemble learning. This method can improve the signal-to-noise ratio during Q-network updates, and stabilize the learning process of the EMDQN agent, which helps the deep Q-network (DQN) [6] operate stably in OTN networks.

The main contributions of this paper are as follows:We propose an SD-OTN routing optimization algorithm based on the reinforcement learning model of the EMDQN. To effectively improve the extrapolation capability of DRL decision-makers, we design a more refined state representation and a limited set of actions.We use the MPNN algorithm instead of the traditional DQN’s policy networks, which can capture the relationship between links and network topology demand and improve the DRL decision-maker performance and generalization capability. Additionally, we exploit the advantages of efficient exploration through ensemble learning to explore the environment in parallel and improve convergence performance.We design practical comparison experiments to verify the superior performance of the EMDQN model.

The rest of this paper is structured as follows. In Section 2, this paper discusses research related to the proposed solution for the network problem. Section 3 describes the software-defined network system architecture and the OTN optimization scenarios and tasks. In Section 4, this paper describes the design of DRL-based routing optimization decisions. In Section 5, this paper presents an extensive evaluation of DRL-based solutions in some realistic OTN scenarios. Finally, in Section 6, we present our conclusion and directions for future work.

## 2. Related Research

Traditional routing optimization schemes are usually based on the OSPF (open shortest path first) [7] or ECMP (equal-cost multipath routing) [8]. The OSPF protocol routes all flow requests individually to the shortest path. The ECMP protocol increases transmission bandwidth using multiple links simultaneously. However, these approaches, based on fixed forwarding rules, are prone to link congestion and cannot meet the demand of exponential traffic growth. Recently, most heuristic algorithm-based approaches have been built under the architecture of the SDN. The authors in [9] proposed a heuristic ant-colony-based dynamic layout algorithm for SDNs with multiple controllers, which can effectively reduce controller-to-switch and controller-to-controller communication delays caused by link failures. The authors in [10] applied a random-based heuristic method called the alienated ant algorithm, which forces ants to spread out across all available paths while searching for food rather than converging on a single path. The authors in [11] analytically extract historical user data through a semi-supervised clustering algorithm for efficient data classification, analysis, and feature extraction. Subsequently, they used a supervised classification algorithm to predict the flow of service demand. The authors in [12] proposed a heuristic algorithm-based solution for DWDM-based OTN network planning. The authors in [13] proposed a least-cost tree heuristic algorithm to solve the OTN path-sharing and load-balancing problem. However, because of a lack of historical experience in data learning, heuristic algorithms can only build models for specific problems. When the network changes, it is difficult to determine the network parameters and there is limited scalability to guarantee service quality. Furthermore, because of the tremendous computational effort and high computational complexity of these methods, heuristic algorithms do not perform well on OTN networks.

With SDN’s maturity and large-scale commercialization, the SD-OTN based on the SDN is becoming increasingly popular in the industry. SD-OTN adapts the reconfigurable optical add-drop multiplexer (ROADM) nodes through the southbound interface protocol and establishes a unified resource and service model. The SD-OTN controller can realize topology and network status data collection, routing policy distribution, and network monitoring. Therefore, many researchers deploy artificial intelligence algorithms in the controller. Deep learning, with its powerful learning algorithms and excellent performance advantages, has gradually been applied to the SDN. To solve the SDN load-balancing problem, Chen et al. [14] used the long short-term memory (LSTM) to predict the network traffic in the SDN application plane. The authors in [15] proposed a weighted Markov prediction model based on mobile user classification to optimize network resources and reduce network congestion. The authors in [16] proposed an intrusion detection system based on SDN and deep learning, reducing the burden of security configuration files on network devices. However, deep learning requires many datasets for training and has poor generalization abilities due to its inability to interact with the environment. These factors make it difficult to optimize the performance of dynamic networks. Compared with deep learning, reinforcement learning uses online learning for model training, changing agent behaviors through continuous exploration, learning, and experimentation to obtain the best return. Therefore, reinforcement learning does not require the model to be trained in advance. It can change its action according to the environment and reward feedback. The authors in [17] designed a Q-learning-based localization-free routing for underwater sensor networks. The authors in [18] proposed a deep Q-routing algorithm to compute the path of any source-destination pair request using a deep Q-network with prioritized experience replay. The authors in [19] proposed traction control ideas to solve the routing problem. The authors in [20] proposed a routing optimization algorithm based on the proximal policy optimization (PPO) model in reinforcement learning. The authors in [21] discussed a solution for automatic routing in the OTN using DRL. Although the studies described above have been successful for the SDN demand-routing optimization problem, they do not perform as well in new topologies because they do not consider the model’s generalization capability.

The traditional DRL algorithms use a typical neural network (NN) as the policy network. The NN can extract and filter the features of the input information and data layer by layer to finally obtain the results of tasks, such as classification and prediction. However, as research advances, conventional neural networks are unable to solve all network routing problems and will struggle to handle non-Euclidean-structured graph data. Therefore, we need to optimize the traditional reinforcement learning algorithm to improve its ability to extract the information features of the sample. Off-policy reinforcement learning (Off-policy RL) algorithms significantly improve sample utilization by reusing past experiences. The authors in [22] propose an off-policy actor–critic RL algorithm based on a maximum entropy reinforcement learning framework. The participants’ goal in this framework is to maximize the expected reward while maximizing the entropy. They achieved state-of-the-art sample efficiency results by combining a maximum entropy framework. However, in practice, the commonly used off-policy approximate dynamic programming methods based on the Q-learning and actor–critic methods are susceptible to data distribution. They can only make limited progress without collecting additional on-policy data. To address this problem, the authors in [23] proposed bootstrap error accumulation reduction to reduce off-policy algorithm instability caused by accumulating backup operators via the Bellman algorithm. The authors in [24] developed a new estimator called offline dual reinforcement learning, which is based on the cross-folding estimation of Q-functions and marginalized density ratios. The authors in [25] used a framework combining imitation learning and deep reinforcement learning, effectively reducing the RL algorithm’s instability. The authors in [26] used the DQN replay datasets to study off-policy RL, effectively reducing the off-policy algorithm’s instability. The authors in [27] proposed an intelligent routing algorithm combining the graph neural network (GNN) and deep deterministic policy gradient (DDPG) in the SDN environment, which can be effectively extended to different network topologies, improving load-balancing capabilities and generalizability. The authors in [28] combined GNN with the DQN algorithm to address the lack of generalization abilities in untrained OTN topologies. OTN topology graphs are non-Euclidean data, and the nodes in their topology graphs typically contain useful feature information that most neural networks are unable to comprehend. They use MPNN to extract feature information between OTN topological nodes, which improves the generalization performance of the DRL algorithm.

However, it is a challenge for a single DRL agent to balance exploration and development, resulting in limited convergence performance. Ensemble learning solves a single prediction problem by building several models. It works by generating several classifiers or models, each of which learns and predicts independently. These predictions are finally combined into a combined prediction, which outperforms any single classification for making predictions [29]. There are two types of integrated base learning machines. One type involves using various learning algorithms on the same dataset to obtain a base learning machine, which is usually referred to as heterogeneous [30,31,32]. The other type applies the same learning algorithm on a different training set (which can be obtained by random sampling based on the original training dataset, etc.), and the base learning machine obtained using this method is said to be a homogeneous type. However, because of the high implementation difficulty and low scalability of heterogeneous types of base learning machines, expansion to high-dimensional state and action spaces is difficult, making it unsuitable for solving OTN routing optimization problems. Table 1 summarizes the description of the papers reviewed, whether SDN and RL are considered, and the evaluation indicators. The EMDQN algorithm we propose applies the same reinforcement learning algorithm to different training sets to generate the base learning machine. We combine multiple EMDQN agents to construct an ensemble learning machine and generate diverse samples to effectively generate learning machines with high generalization abilities and significant differences.

## 3. SD-OTN Architecture

In this paper, the designed SD-OTN architecture consists of the application, control, and data planes, as shown in Figure 1. The description of each part of the network architecture is as follows:

Data plane. The data plane consists of the ROADM nodes and the predefined optical paths connecting them. In the data plane, the capacity of the links and the connection status of the ROADM nodes are predefined. The data plane must collect the current optical data unit (ODU) signal requests and network status information, which it must then send to the control plane via the southbound interface. The data plane implements the new routing forwarding policy after receiving it from the control plane. It communicates the new network state and traffic demand to the control plane, from which decision-makers in the application plane learn.Control plane. The control plane consists of the SDN controller. The control plane obtains the ODU signal request and network status information via the southbound interface and calculates the reward using the reward function. Through the northbound interface, the control plane sends the network state, traffic demand, and reward to the application plane via the northbound interface. When receiving optimized routing action from the application plane, the control plane sends a routing forwarding policy to the data plane based on the routing action.Application plane. The application plane manages the EMDQN agents. The agents obtain network state information from the control plane, encode it, and feed it into the agents’ policy network, which generates optimized routing actions. Subsequently, the routing actions are sent down to the control plane.

## 4. EMDQN-Based Decision Design for Routing Optimization

In this section, we describe in detail the EMDQN algorithm proposed in this paper.

### 4.1. DRL-Based Routing Optimization in SD-OTN

Based on the system architecture described above, the DRL agent’s role is to assign routes to incoming traffic demands for a specific sequence of optical paths (i.e., end-to-end paths) to maximize network utility. Because the DRL agent operates in the electrical domain, traffic demands are treated as requests for ODU signals. These signals, which may originate from different clients, are multiplexed into an optical transform unit (OTU), as shown in Figure 1. The final OTU frames are transmitted through the optical channels in the OTN [33].

We use G to refer to an optical transmission network, as shown in Equation (1):(1)G=(V,E)
where V and E represent the set of n ROADM nodes and m optical links in the network topology, respectively, as shown in Equations (2) and (3).
(2)$$V=\left[v_{1},\;v_{2},\ldots ,\;v_{n}\right].$$
(3)$$E=\left[e_{1},e_{2},\ldots ,e_{m}\right].$$

We use C to denote the set of link bandwidth capacity, as shown in Equation (4), where |C|=|E|=m:(4)$$C=\left[c_{1},c_{2},\ldots ,c_{m}\right].$$

The path k from node $v_{i}$ to node $v_{j}$ is defined as a sequence of links, as shown in Equation (5), where $e_{k\left(i\right)}\in E$:(5)$$p_{k}=\left\{e_{k\left(0\right)},e_{k\left(1\right)},\ldots ,e_{k\left(n\right)}\right\}.$$

We use $d_{k}$ to denote the traffic demand of the path k, and define D as the set of all traffic demands, as shown in Equation (6):(6)$$D=\left[d_{1},d_{2},\ldots ,d_{n*n}\right].$$

The traffic routing problem in OTN is a classical resource allocation problem [26]. If the bandwidth capacity of the distributed routing path is greater than the size of the bandwidth requirement, the allocation is successful. After successfully allocating bandwidth capacity for a node pair’s traffic demand, the routing path will not be able to release the bandwidth occupied by that demand until the end of this episode. We use $rb_{i}$ to describe the remaining bandwidth of the link $e_{i}$, which is the link bandwidth capacity $c_{i}$ minus the traffic demands of all paths passing through link $e_{i}$, as shown in Equation (7). RB is the set of the remaining bandwidth of all links, as shown in Equation (8).
(7)$$rb_{i}=c_{i}-\sum d_{k}.$$
(8)$$RB=\left[rb_{1},rb_{2},\ldots ,rb_{m}\right].$$

We use $q_{k}$ to denote the allocating traffic demand of the path k, as shown in Equation (9). Q is the set of all allocating traffic demands, as shown in Equation (10).
(9)$$q_{k}=\{\begin{array}{c} d_{k},\;\;if\;\forall e\in p_{k}\;and\;r_{e}>d_{k}\;\\ 0,\;\;else \end{array}.$$
(10)$$Q=\left[q_{1},q_{2},\ldots ,q_{n*n}\right].$$

The optimization objective in this paper is to successfully allocate as much of the traffic demand as possible, as shown in Equation (11):(11)$$\max\left(\sum_{q_{i}\in Q}q_{i}\right).$$

In view of the above optimization objective, the routing optimization can be modeled as a Markov decision process, defined by the tuple {S, A, P, R}, where S is the state space, A is the action space, P is the set of transfer probabilities, and R is the set of rewards. The specific design is as follows:Action space: The action space is designed as k shortest hop-based paths of source-destination nodes. The action selects one of the k paths to transmit the traffic demand of source–destination nodes. The parameter *k* is customizable and varies according to the topology’s complexity. The action space is invariant to the arrangement of nodes and edges, which is discretely distributed, allowing the DRL agent to understand the actions on arbitrary network states easily.State space: The state space is designed as the remaining bandwidth RB, the traffic demand D, and the link betweenness. The link betweenness is a centrality metric, which indicates how many paths are likely to cross the link. For each node pair in the topology, k candidate shortest routes are calculated, with the link betweenness value being the number of shortest routes passing through the link divided by the total number of paths, as shown in Equation (12), where $bn_{i}$ represents the betweenness of the link $e_{i}$, N represents he total number of paths, $p_{i}^{k}$ represents the number of shortest routes passing through the link $e_{i}$ in k candidate shortest routes:
(12)$$bn_{i}=p_{i}^{k}/N.\;$$
Reward function: The reward function returns a positive reward if the selected link has sufficient capacity to support the traffic demand in an episode; otherwise, it returns no reward and terminates the episode. According to the optimization objective in Equation (11), the final reward for the episode is the sum of the rewards of all successfully assigned traffic demand tuples {src, dst, demand}, as shown in Equation (13), where N is the number of traffic demand tuples, $r_{t}$ represents the reward after the action at time t, $q_{i}$ represents the i-th traffic demand successfully assigned, and $q_{max}$ represents the maximum traffic demand successfully assigned. The higher the reward, the more bandwidth demands are successfully allocated in that time step, and the better the network load-balancing capability..


(13)
$$r_{t}=\sum_{i=1}^{N}q_{i}/q_{max}.$$


### 4.2. DQN Algorithm

Based on the above DRL-based optimization solution, this paper selects the DQN algorithm to implement a reinforcement learning agent. The DQN is a classical DRL algorithm based on value functions. It combines a convolutional neural network (CNN) with the Q-learning algorithm, using the CNN model to output the Q-value corresponding to each action to ascertain which to perform [6].

The DQN algorithm uses two network models containing CNNs for learning: the prediction network Q(s,a,θ) and the target network$\hat{\;Q}\left(s,a,\bar{\theta }\right)$, where θ and $\bar{\theta }$ are the network parameters of the prediction and target networks, respectively. The prediction network outputs the predicted Q-value corresponding to the action, whereas the target network calculates the target value and updates the parameters of the prediction network based on a loss function. The DQN copies the parameters of the prediction network model to the target network after each C-round iteration.

The prediction network approximates the action value function through the CNN model $Q_{\pi }\left(s,a\right)$, as shown in Equation (14):(14)$$Q\left(s,a,\theta \right)\approx Q_{\pi }\left(s,a\right).$$

The DQN agent selects and executes an action based on an ϵ-greedy policy. The policy generates a random number in [0, 1] interval through a uniform distribution. If the number is less than 1−ϵ, it selects an action that maximizes the Q-value; otherwise, it selects an action randomly, as shown in Equation (15):(15)$$a_{t}=\{\begin{array}{ll} \arg\underset{a}{\max}\;Q\left(s_{t},a,\theta \right), & \mathrm{with}\;\mathrm{probability}\;1-\varepsilon \\ \mathrm{random}\;\mathrm{action}, & \mathrm{otherwise} \end{array}.$$

The target network calculates the target value y by obtaining a random mini-batch storage sample from the replay buffer, as shown in Equation (16), where r is the reward value and γ is the discount factor:(16)$$y=r+\gamma \underset{a^{\prime }}{\max}\hat{Q}\left(s^{\prime },a^{\prime },\bar{\theta }\right).$$

The DQN defines the loss function of the network using the mean-square error, as shown in Equation (17). The parameter θ is updated by the mini-batch semi-gradient descent, as shown in Equations (18) and (19):(17)$$L\left(\theta \right)=E\left[{\left(y-Q\left(s,a,\theta \right)\right)}^{2}\right],$$
(18)$$\nabla_{\theta }L\left(\theta \right)\approx \frac{1}{N}\sum_{i}^{N}\left(y-Q\left(s,a,\theta \right)\right)\nabla Q\left(s,a,\theta \right),$$
(19)$$\theta \leftarrow \theta -\alpha \nabla_{\theta }L\left(\theta \right),$$
where N represents the number of samples and α represents the update parameter.

The target network is used by the DQN to keep the target Q-value constant over time, which reduces the correlation between the predicted and target Q-values to a certain extent. This operation reduces the possibility of loss value oscillation and divergence during training and improves the algorithm’s stability.

### 4.3. Message-Passing Neural Network

The CNN model has better results in extracting features from Euclidean spatial data (e.g., picture data), characterized by a stable structure and dimensionality. However, graph-structured or topologically structured data are infinitely dimensional and irregular, and the network surrounding each node may be unique. Such structured data renders traditional CNNs ineffective and unable to extract data features effectively. To address this problem, we use the MPNN rather than the CNN as a network model for the DQN. The MPNN is a type of GNN that is suitable for extracting spatial features of topological graph data [5].

Through repeated iterations of the process of passing data about the link’s hidden state, the MPNN algorithm abstract information about the characteristics of the network. The characteristic values of the hidden state $h_{i}$ include the remaining bandwidth $rb_{i}$, the link betweenness $bn_{i}$, and the traffic demand feature $df_{i}$. The traffic demand feature $df_{i}$ represents the quantitative characteristics of the traffic demand $d_{i}$. Because the traffic demand of the OTN environment is discrete and finite, the traffic demand feature is denoted by an n-element one-hot encoding, and link characteristics that are not included in the k routes have a zero value. Additionally, the size of the hidden state is usually larger than the size of the feature values in the hidden state; thus, we use zero values to populate the feature value vector, as shown in Equation (20):(20)$$h_{i}=\left[rb_{i},bn_{i},df_{i},0,\ldots ,0\right].$$

The MPNN workflow is shown in Figure 2. We perform a message-passing process between all links which will be executed T times. First, the MPNN receives link hidden features as the input. Second, each link iterates over all of its adjacent links to obtain the link features. In the message-passing process, for each link k, we generate messages by entering the hidden state $h_{k}$ of the link and the hidden state $h_{i}$ of all neighboring links into the message function m(·). The message function m(·) is a fully connected CNN. After iterating over all links, the link k receives messages from all neighboring links (denoted by N(k)). It generates a new feature vector $M_{k}$ using message aggregation, as shown in Equation (21):(21)$$M_{k}^{t+1}=\sum_{i\in N\left(k\right)}m\left(h_{k}^{t},h_{i}^{t}\right),$$
where N(k) represents all neighboring links of the link k.

Second, we update the hidden state of the link by aggregating the feature vector $M_{k}^{t+1}$ with the link-hidden state $h_{k}^{t}$ through the update function u(·), as shown in Equation (22). The update function u(·) is the Gate Recurrent Unit (GRU), which has the characteristics of high training efficiency.
(22)$$h_{k}^{t+1}=u\left(h_{k}^{t},M_{k}^{t+1}\right).$$

Finally, after the T-step message transmission, we use the readout function R(·) to aggregate the hidden state of all links and obtain the Q-value, as shown in Equation (23):(23)$$Q\left(s,a,\theta \right)=R\left(\sum_{k\in E}h_{k}\;\right),$$
where E represents the set of all links in the topology.

### 4.4. Ensemble Learning

In the DQN algorithm, it is challenging for a single agent to balance exploration and development, resulting in limited convergence performance. Furthermore, errors in the DQN target values can increase the overall error in the Q-function, leading to an unstable convergence. In this paper, we use ensemble learning to solve the above problems. Ensemble learning has the advantage of efficient exploration and can reduce uncertainty in new samples.

As shown in Figure 3, ensemble learning is realized by a set of multiple EMDQN agents ${\left\{Q\left(s,a,\theta_{i}\right)\right\}}_{i=1}^{N}$, where $\theta_{i}$ represents the parameter of the i-th agent. To diversify the training of the EMDQN agents, we randomly initialize the policy network of all EMDQN agents. In the training phase, we employ the *ϵ*-greedy-based upper-confidence bound (UCB) exploration strategy [34], as shown in Equation (24):(24)$$a_{t}=\{\begin{array}{ll} \underset{a}{\max}\left\{Q_{\mathrm{mean}}\left(s_{t},a,\theta \right)+\lambda Q_{\mathrm{std}}\left(s_{t},a,\theta \right)\right\}, & \mathrm{with}\;\mathrm{probability}\;1-\varepsilon \\ \mathrm{random}\;\mathrm{action}, & \mathrm{otherwise} \end{array},$$
where $Q_{\mathrm{mean}}\left(s_{t},a,\theta \right)$ and $Q_{\mathrm{std}}\left(s_{t},a,\theta \right)$ are the mean and standard deviation of the Q-values output by all MPNN policy networks ${\left\{Q\left(s,a,\theta_{i}\right)\right\}}_{i=1}^{N}$. The exploration reward λ>0 is a hyper-parameter. When λ increases, the EMDQN agents become more active in accessing unknown state–action pairs.

The traditional DQN loss function (Equation (6)) may be affected by error propagation, that is, it propagates the target Q-network $\hat{Q}\left(s^{\prime },a^{\prime },\bar{\theta }\right)$ error to the current state of the Q-network Q(s,a,θ). This error propagation can lead to an unstable convergence. To alleviate this problem, for each EMDQN agent i, this paper uses Bellman weighted backups, as shown in Equation (25):(25)$$L_{WQ}^{\mathrm{EMDQN}}\left(\theta_{i}\right)=w\left(s\right){\left(r+\gamma \underset{a^{\prime }}{\max}\hat{Q}\left(s^{\prime },a^{\prime },\bar{\theta_{i}}\right)-Q\left(s,a,\theta_{i}\right)\right)}^{2},$$
where w(s) represents the confidence weight of the set of target Q-networks in the interval [0.5, 1.0]. w(s) is calculated from Equation (26), where the weight parameter W is a hyper-parameter, σ is a sigmoid function, ${\hat{Q}}_{std}\left(s\right)$ is the empirical standard deviation of all target Q-networks ${\left\{\underset{a}{\max}\hat{Q}\left(s,a,\bar{\theta }\right)\right\}}_{i=1}^{N}$. $L_{WQ}^{EMDQN}\left(\cdot \right)$ reduces the weights of sample transitions with high variance between target Q-networks, resulting in better signal-to-noise ratios for network updates.
(26)$$w\left(s\right)=\sigma \left(-{\hat{Q}}_{std}\left(s\right)*W\right)+0.5.$$

### 4.5. The Working Process of the EMDQN Agent

The working process of the EMDQN agent at each iteration is described in Algorithm 1. We first reset the environment and obtain the environment link capacity and traffic demand tuple {src, dst, demand} (line 1). Subsequently, we execute a loop to continuously assign traffic demands. In the process, we compute k shortest links (Line 3) and allocate the traffic demand for each shortest link through k cycles (Lines 4–8). Based on this, we can compute the Q-value for each action. We select actions using an ϵ-greedy-based UCB exploration strategy (Line 9); subsequently, we apply the chosen route to the environment (Line 10). We store the rewards and state transfer during the interaction with the environment in the experience replay buffer (Line 11) while applying the transferred state (Line 12). The cycle stops when any link is unable to carry the traffic demand. Next, we execute the agent learning phase. For the sampled batch (Line 15), we plot the mask using the Bernoulli distribution (Line 16) and calculate the batch weight using all EMDQN agents. Following that, we multiply the sample by the weight and mask to minimize $L_{WQ}^{EMDQN}$ (Line 18). Finally, we evaluate the set of EMDQN agents in the environment (Line 21) and collect the rewards, as well as the status of the environment, in the evaluation process to analyze the training situation of the EMDQN agent.
**Algorithm 1:** working process of the EMDQN agent1: *s, demand, src, dst* ← env.reset() 2: while (Done != False) do 3:  *k_path* ← compute_k_path(*k, src, dst*) 4:  for *i* ← 1 to *k* do 5:   path ← get_path(*i, k_path*) 6:   *s’* ← allocate(*s, path, src, dst, demand*) 7:   k_Q[*i*] ← compute_Q(*s’, path*) 8:  end for 9:  a ← act(*k_Q, ε, k_path, s*) 10:  *s’, r, done, demand’, src’, dst’* ← env.step(*s, a*) 11:  agent.rememble(*s, a, r, s’, done*) 12:  *s, demand, src, dst* ← *s’, demand’, src’, dst’*13: end while 14: for *i* ← 1 to STEP do 15:  batch ← sample() 16:  m ← bernoulli() 17:  for each agent *i* do 18:   Update agent by minimizing $L_{\mathrm{EMDQN}}^{\mathrm{WQ}}(\theta_{i})$19:  end for 20: end for 21: agent.evaluate() 

## 5. Experiments and Analysis

In this section, we simulate the SD-OTN routing scenario using the OpenAI gym framework to train and evaluate the EMDQN algorithm. Furthermore, we conduct experiments and analyses by adjusting the hyper-parameters and evaluating the algorithm load-balancing ability and generalization ability.

### 5.1. Experimental Environment

The computer used for the experiments has an AMD R5 5600G processor with a base frequency of 2900 MHz, a 2 TB solid-state drive, and 32 GB of RAM. The experiment uses the Tensorflow deep learning framework to implement the EMDQN algorithm. We select NSFNET, GEANT2, and GBN for the optical transmission network topology, with the lightpath bandwidth being 200 ODU0 bandwidth units, as shown in Figure 4. Among these, the NSFNET network contains 14 ROADM nodes and 21 lightpaths, the GEANT2 network contains 24 ROADM nodes and 36 lightpaths, and the GBN network contains 17 ROADM nodes and 27 lightpaths.

In this paper, the lightpath bandwidth requirements are expressed as multiples of ODU0 signals, i.e., 8, 32, and 64 ODU0 bandwidth units. In each episode, the environment generates a traffic demand tuple {src, dst, demand} at random. Additionally, the EMDQN agent should assign the appropriate route for each tuple received. If the assignment is successful, it will receive a reward as defined in Equation (1). Otherwise, it will not be rewarded. Since the new traffic demand is randomly generated, the routing policy designed by the EMDQN agent does not rely on traffic demand distribution information, reducing the EMDQN agent’s overfitting to the particular network scenario used for training.

### 5.2. Hyper-Parameters Settings

We experimentally select suitable hyper-parameters for the EMDQN agent, as shown in Figure 5. In the experiments, we chose the NSFNET as the experimental network topology. The size of the link-hidden state is related to the amount of coding information. We set the size of the link-hidden state to twenty and the number of feature values to five, and filled the rest with zero. To facilitate observation, we smoothed the data when drawing the graph.

Figure 5a shows the training results for the different numbers of EMDQN agents. When the number of agents is high, the training slows down and aggravates the overfitting of DRL agents in the application scenario, resulting in poorer results. The performance is optimal when the number of EMDQN agents is two. Figure 5b shows the training results of the stochastic gradient descent algorithm with different learning rates. When the learning rate was 0.001, the algorithm reward achieved the highest value. Figure 5c shows the training results for different decay rates of ϵ. In the initial stage of training, ϵ is close to 1. We executed 70 iterations and started to reduce ϵ exponentially using ϵ-decay until it decreased to 0.05. During the process of ϵ reduction, the training curve tends to flatten out, finally reaching convergence. The training results show that the reward value curve is most stable after convergence when ϵ-decay is 0.995. Figure 5d shows the training results for different λ values in Equation (11). λ denotes the exploration reward of the EMDQN agent. From the results in Figure 5d, it is clear that the algorithm reward value is highest when λ value is 5. Figure 5e depicts the training results for different weight parameters W in Equation (14). In this paper, we set the size of samples to 32. As W increases, the sample weights converge and become less than one, which affects the sample efficiency of the EMDQN. The reward of this algorithm reaches its highest value when the value of W is 0.05.

Table 2 shows some relevant parameters of the EMDQN and values taken after tuning the parameters.

### 5.3. Load-Balancing Performance Evaluation

In this section, we experimentally evaluate the EMDQN in the three network topologies, as described in Section 5.1. The DRL agent runs 2000 iterations. In each iteration, the agent trains 50 episodes and evaluates 40 episodes. Furthermore, the DRL agent updates the network during the training period. During the evaluation, the DRL agent does not update the network; rather, it applies the action to the environment intending to maximize the Q-function, and subsequently records network state data, such as rewards, link utilization, and throughput for each episode.

We implement other SDN solutions for performance comparison with EMDQN algorithms, such as OSPF [7], ECMP [8], DQN [17], PPO [19], and DQN+GNN [26]. The DQN+GNN is an ablation experiment among the compared algorithms, i.e., a performance comparison of the EMDQN model with ensemble learning removed. The DQN and PPO are classic DRL algorithms that use a fully connected feedforward NN as a policy network. The OSPF is an open shortest path algorithm that performs an action selection by calculating the shortest number of hops of the link traversed between the source and destination nodes. The ECMP algorithm is an equal-value multipath routing protocol that allows the use of multiple links simultaneously in the network. The ECMP algorithm distributes the bandwidth demand equally over k lightpaths in this experiment. Furthermore, OUD0 signals are not divisible, but we can verify the performance in other network scenarios in this way.

Figure 6 shows the average reward of all algorithms for the three evaluation scenarios, where the confidence interval is 95%. In this paper, we design the reward based on whether the bandwidth demand can be successfully allocated. The greater the reward, the more bandwidth demand is successfully allocated, and the better the network load-balancing capability. In all three evaluation scenarios, the EMDQN algorithm proposed in this paper performs better than other algorithms after convergence. The EMDQN algorithm outperforms the DQN+GNN algorithm with ensemble learning removed after convergence by more than 7%, demonstrating that the multi-agent ensemble learning approach can effectively improve the convergence performance of the DQN. Additionally, the EMDQN and DQN+GNN outperform the classical reinforcement learning algorithms (DQN and PPO) by more than 25% in all three evaluated scenarios. This indicates that the MPNN can effectively improve the decision performance of the reinforcement learning model by capturing information about the relationship between the demand on links and network topology. The DQN and PPO algorithms perform about as well as the ECMP algorithm after convergence. The OSPF algorithm, on the other hand, routes all flow requests singularly to the shortest path. Since this method is based on fixed forwarding rules, it can easily lead to link congestion. Therefore, the OSPF algorithm is only close to ECMP in the GBN scenario and the lowest in other scenarios.

Table 3 shows the average throughput of each algorithm in ODU0 bandwidth units for the three network topologies. Table 4 displays the average link utilization of each algorithm across the three network topologies. The average throughput and link utilization of the EMDQN are higher than those of other algorithms under various network topologies, indicating that the EMDQN algorithm has a better load-balancing capability for the network after convergence. The performance of the EMDQN algorithm is higher than that of the DQN+GNN algorithm, which is a good indication that ensemble learning can improve the convergence performance of the model. The results show that the EMDQN has excellent decision-making abilities.

### 5.4. Generalization Performance Evaluation

In a real OTN scenario, there is the possibility that the network topology changes due to a broken lightpath. In this case, the DRL model usually needs to be retrained, resulting in a network state that is low-load-balanced for an extended period, which is intolerable for real network situations. To verify the generalization performance of the EMDQN in this paper, we simulated the light path breakage case in the training environment. We randomly break 0–10 lightpaths in each network scenario and evaluate 100 iterations using the converged EMDQN model while ensuring that the network topology remains connected.

In the generalization experiments, we compared and analyzed the EMDQN, DQN, and OSPF. The classical DQN algorithm is implemented using a fully connected network and will fail if the network topology changes. To avoid retraining the DQN model, we removed some network parameters and applied the same evaluation method after adjusting the state inputs. Figure 7 shows the experimental results of the model’s evaluation of randomly malfunctioning lightpaths in different network scenarios. When a lightpath malfunctions, a new route needs to be found to avoid the failed lightpath. As the number of faulty lightpaths increases, fewer routes become available, resulting in a reduction in network transmission traffic and a decrease in the load capacity of the network. The OSPF and the classical DQN algorithms have progressively lower rewards as the number of faulty lightpaths increases and have worse performance than the EMDQN model. In contrast, the EMDQN algorithm can still understand the state of the network and obtain a higher reward. The results demonstrate that the MPNN can still improve the model’s generalization ability in the case of network failure, which allows the EMDQN agent to maintain a good performance.

To further verify the generalization performance of EMDQN agents, we use the EMDQN model trained to converge in NSFNET to transfer to GEANT2 and GBN network topologies for evaluation. Because of the generalization capabilities of the MPNN, the converged EMDQN agents can directly transfer to operate in different network topologies. The experimental results are shown in Figure 8. The classical DQN algorithm does not perform effectively as the traditional routing algorithm OSPF when the network topology is changed. However, the EMDQN model in this paper still works stably, and the reward section, average value, and stability are significantly better than the DQN and OSPF algorithms. This confirms that the EMDQN agent can still maintain excellent decision-making abilities in the case of network connectivity changes.

## 6. Conclusions and Future Work

In this paper, we proposed the EMDQN algorithm, which uses the MPNN as a policy network for the DRL to improve the DRL agent’s decision-making and generalization abilities, allowing the EMDQN agent to efficiently generalize the unknown topology. We verify the convergence and generalization performance of the EMDQN algorithm by SD-OTN simulating experiments, analyzing and comparing traditional routing protocols with some SDN solutions based on other DRL algorithms. The experimental results show that the EMDQN model can generalize unknown network topologies and outperform other SDN solutions. Furthermore, integrating multiple agents using integrated learning and combining weighted Bellman backup as well as UCB exploration strategies improves the convergence performance while alleviating the DRL agents’ unstable operation when converging.

However, the difficulty of adjusting parameters is one challenge faced by the DRL. As seen in Section 5.3, the EMDQN has more hyper-parameters and requires several experiments to complete the adjustment parameters. Therefore, in our future work, we will continue to improve the DRL algorithm and reduce the parameter sensitivity of the DRL method to reduce the reality gap in the DRL method’s landing.

## Figures and Tables

**Figure 1 sensors-22-08139-f001:**
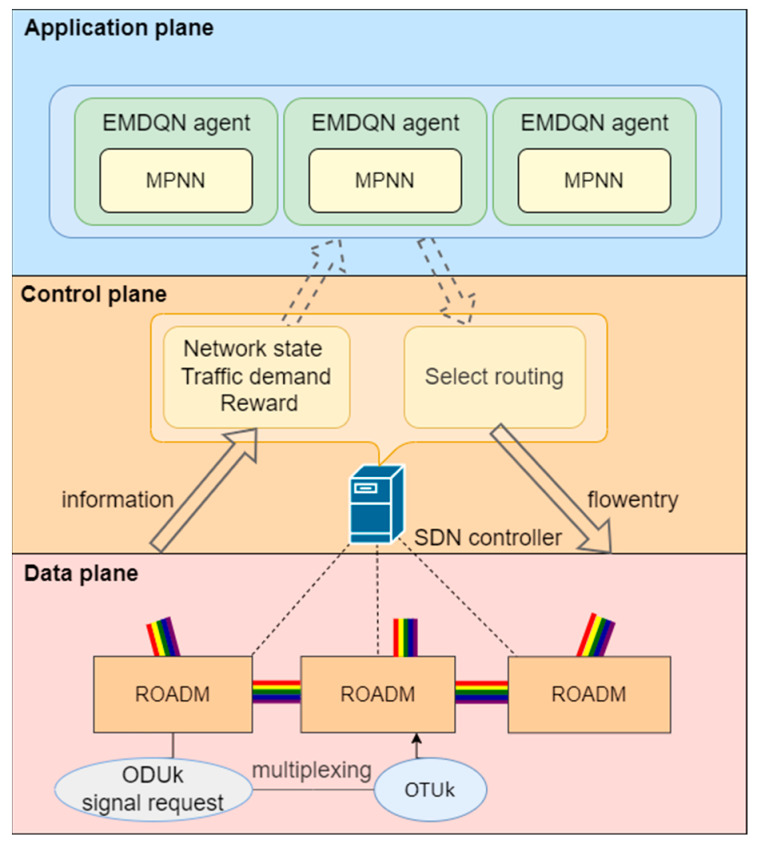
The SD-OTN architecture. The architecture consists of the application plane, control plane, and data plane.

**Figure 2 sensors-22-08139-f002:**
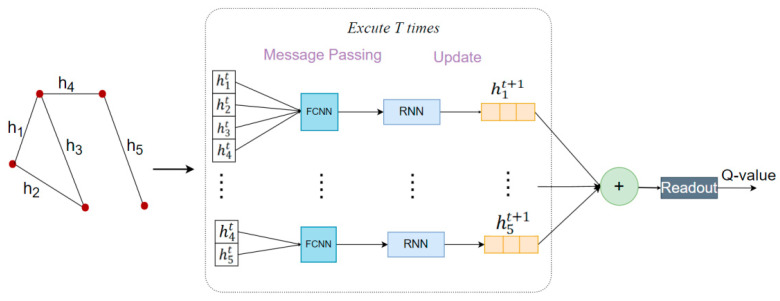
The MPNN workflow.

**Figure 3 sensors-22-08139-f003:**
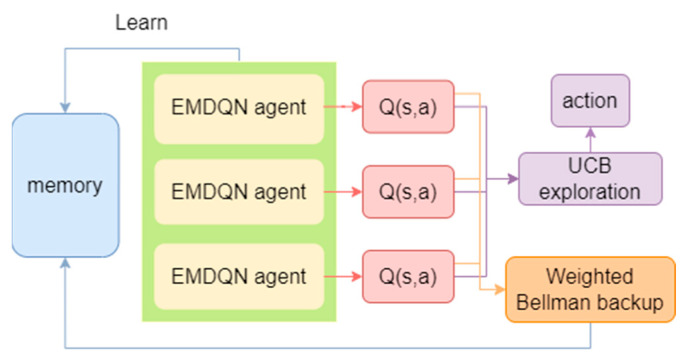
EMDQN workflow.

**Figure 4 sensors-22-08139-f004:**
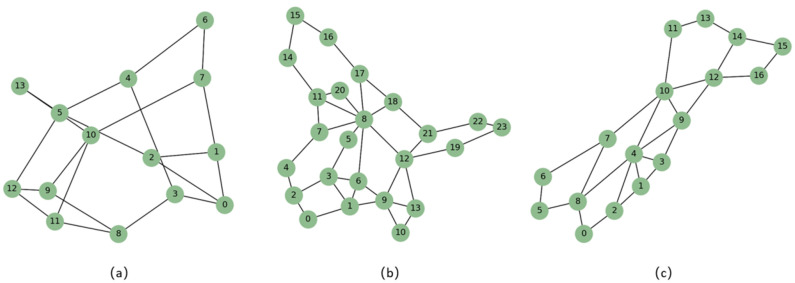
The optical transmission network topologies: (**a**) the NSFNET topology; (**b**) the GEANT2 topology; (**c**) the GBN topology.

**Figure 5 sensors-22-08139-f005:**
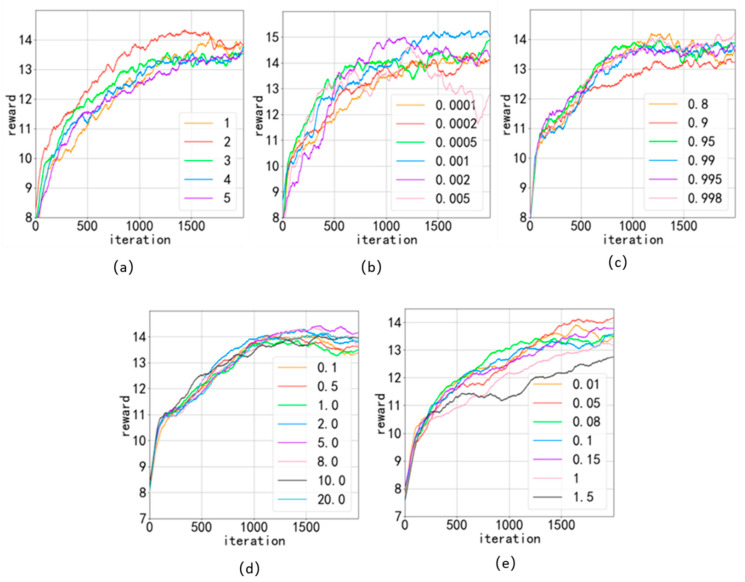
Comparison of the effect of some super-references: (**a**) ensemble number; (**b**) learning rate; (**c)** ϵ-decay; (**d**) UCB exploration reward λ; (**e**) weight parameter W.

**Figure 6 sensors-22-08139-f006:**
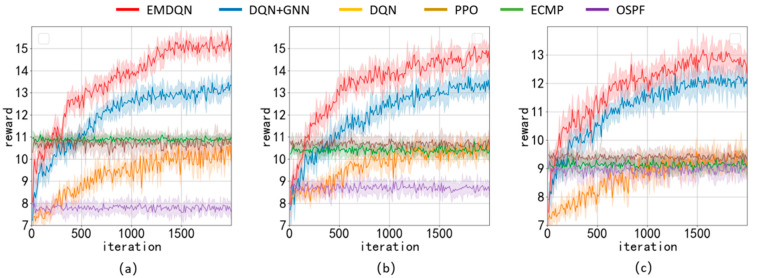
Comparison of the rewards of each algorithm in different scenarios: (**a**) NSFNET scenario evaluation; (**b**) GEANT2 scenario evaluation; (**c**) GBN scenario evaluation.

**Figure 7 sensors-22-08139-f007:**
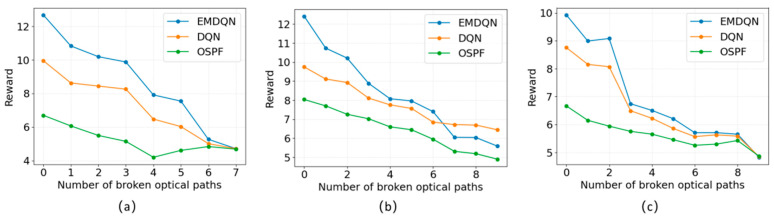
Evaluation of the model in different network scenarios with randomly broken lightpaths: (**a**) broken lightpath at NSFNET; (**b**) broken lightpath at GEANT2; (**c**) broken lightpath at GBN.

**Figure 8 sensors-22-08139-f008:**
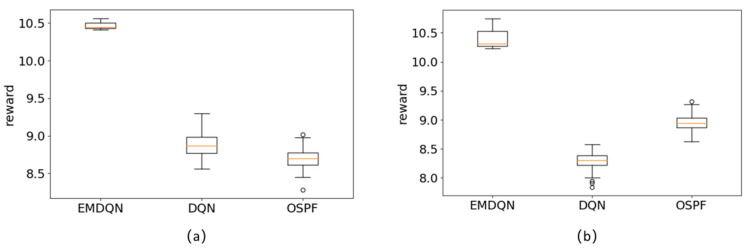
Performance of the algorithm after changing the network topology: (**a**) train in the NSFNET and evaluate in the GEANT2; (**b**) train in the NSFNET and evaluate in the GBN.

**Table 1 sensors-22-08139-t001:** Related work.

Paper	Description	RL	DL	OTN	Evaluating Indicator
[7]	Performance analysis of OSPF				Network convergence, traffic dropped
[8]	Embarks upon a systematic algorithmic study of traffic engineering with ECMP				Throughput
[9]	Allocation of computational resources based on heuristic ant colony algorithm				Latency, load balancing, task completion time.
[10]	A load-balancing algorithm based on the alienated ant algorithm				Throughput, delay, packet loss rate
[11]	SDN routing solution about flow feature extraction, requirement prediction and route selection				Routing efficiency
[12]	An OTN network planning solution over DWDM based on heuristic algorithms			√	Network resource consumption
[13]	A heuristic algorithm of minimum cost tree for path sharing and load balancing			√	Tree cost, run time, degree of load balancing
[14]	A network traffic prediction model based on LSTM		√		Throughput, load-balancing degree
[16]	Deep learning classifier for detection of anomalies		√		Precision, recall, accuracy of classification
[17]	A Q-learning-based localization-free anypath routing	√			Delay, network lifetime, packet delivery ratio
[19]	Combines the control theory and DRL technology to achieve an efficient network control scheme	√	√		Transmission delay
[20]	An RL routing algorithm to solve a traffic engineering	√	√		Throughput and delay, transmission time
[21]	Designing state and action to simplify the DRL algorithm	√	√	√	Link utilization
[27]	A set of extensions to the MQTT protocol that meet application-defined real-time requirements	√	√		Latency
[28]	A DRL algorithm combined with GNN	√	√	√	Network capacity
[30]	A new method of data missing estimation with tensor heterogeneous ensemble learning		√		Data missing rates
[32]	A method to automatically learn long-term associations between traffic samples		√		Calculates precision, recall and F1-score

**Table 2 sensors-22-08139-t002:** Some relevant parameters of the EMDQN.

Parameter	Value
Batch size	32
Learning rate	0.001
Soft wights copy α	0.08
Dropout rate	0.01
State hidden	20
Ensemble number	2
UCB exploration reward λ	5
Weight parameter W	0.05
ϵ-decay	0.995
Discount factor γ	0.95

**Table 3 sensors-22-08139-t003:** A comparison of the average throughput of each algorithm in different scenarios.

	EMDQN	DQN+GNN	DQN	PPO	ECMP	OSPF
NSFNET	**1028.17 ± 27.45**	899.28 ± 24.21	709.47 ± 35.47	737.48 ± 26.92	747.68 ± 13.50	548.86 ± 20.97
GEANT2	**995.56 ± 35.26**	903.48 ± 35.36	721.97 ± 31.07	726.89 ± 24.09	717.35 ± 19.59	605.27 ± 21.53
GBN	**864.77 ± 30.28**	826.06 ± 30.19	646.56 ± 29.71	652.05 ± 15.51	636.10 ± 18.69	627.70 ± 23.02

**Table 4 sensors-22-08139-t004:** A comparison of the average link utilization of each algorithm in different scenarios.

	EMDQN	DQN+GNN	DQN	PPO	ECMP	OSPF
NSFNET	**56 ± 0.90%**	50 ± 1.28%	40 ± 1.69%	43 ± 0.65%	41 ± 1.04%	15 ± 1.16%
GEANT2	**40 ± 0.92%**	36 ± 1.46%	29 ± 1.49%	30 ± 0.60%	21 ± 0.90%	11 ± 0.83%
GBN	**45 ± 0.92%**	42 ± 1.54%	34 ± 1.48%	35 ± 0.76%	25 ± 1.22%	15 ± 1.07%

## Data Availability

The authors confirm that the data supporting the findings of this study are available within the article.

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
