# Peer review of "A Routing Optimization Method for Software-Defined Optical Transport Networks Based on Ensembles and Reinforcement Learning"

_sensors, 2022, doi:10.3390/s22218139_

Round 1

Reviewer 1 Report

The title of this article is acceptable.

This article is related to the EMDQN algorithm, which uses the MPNN as a policy network for the DRL to improve the DRL agent’s decision-making and generalization abilities. It allows the EMDQN agent to efficiently generalize the unknown topology.

Based on experimental results show that the EMDQN model can generalize unknown network topologies and outperform other SDN solutions. Besides, integrating multiple agents using integrated learning and combining weighted Bellman backup as well as UCB exploration strategies improves the convergence performance while alleviating the DRL agents’ unstable operation when converging.

However, the challenge faced by the DRL is that the EMDQN has more hyper-parameters and requires several experiments to complete the adjustment parameters. 

Overall this article can be accepted but authors required to do minor correction as stated below: 

1. Figure 2 need to be referred in this article. Please confirm the statement related to Figure 2.

2. Is better if author can adjust or rearrange Figure 2, 4, 5, 6, 7 and 8.

References is good and referred to latest journal.

Author Response

Point 1: Figure 2 need to be referred in this article. Please confirm the statement related to Figure 2. 

 Response 1: We apologize for our careless mistakes. Thank you for your reminder. We correct “Figure 3” in Line 346 to “Figure 2” and supplement the relevant description of Figure 2 in Line 346-349. (“The MPNN workflow is shown in Figure 2. We perform a message passing process between all links which will be executed T times. First, the MPNN receives link hidden features as input. Second, each link iterates over all of its adjacent links to obtain the link features.”)

Point 2: Is better if author can adjust or rearrange Figure 2, 4, 5, 6, 7 and 8.

Response 2: Thank you for your significant suggestion. We rearrange Figure 2-8 in the new manuscript.

Reviewer 2 Report

The manuscript proposes an Ensembles and Message passing neural network-based Deep Q-Network (EMDQN) method for optical network routing optimization to adapt to the sudden changes in the network and improve the sample usage. The following points are observed. 

1. The manuscript is well organized and structured.

2. The related work section seems a bit weak. It is suggested to add a tabular comparison of the existing approaches on the topic and identifying the research gaps. 

3. The methodology and results are clearly presented.  

Author Response

Point 1: The related work section seems a bit weak. It is suggested to add a tabular comparison of the existing approaches on the topic and identifying the research gaps.

Response 1: Thank you for this suggestion. We updated the manuscript by adding additional remarkable references and enlarged the review. New references [16] [25] regarding deep learning and reinforcement learning and corresponding reviews were added to the resubmitted manuscript. In addition, we add Table 1 in Section II to summarize the description of the papers reviewed, whether SDN and RL are considered, and the evaluation indicators.
